# Participation of Oxidative Stress in the Activity of Compounds Isolated from *Eleutherine plicata* Herb

**DOI:** 10.3390/molecules28145557

**Published:** 2023-07-20

**Authors:** Antônio Rafael Quadros Gomes, Jorddy Neves Cruz, Ana Laura Gadelha Castro, Heliton Patrick Cordovil Brigido, Everton Luiz Pompeu Varela, Valdicley Vieira Vale, Liliane Almeida Carneiro, Gleison Gonçalves Ferreira, Sandro Percario, Maria Fâni Dolabela

**Affiliations:** 1Postgraduate Program in Pharmaceutical Innovation, Institute of Health Sciences, Federal University of Para, Belem 66075-110, PA, Brazil; rafaelquadros13@hotmail.com (A.R.Q.G.); lauracastro.farmacia@gmail.com (A.L.G.C.); helitom2009@hotmail.com (H.P.C.B.); valdicleyvale@gmail.com (V.V.V.); 2Oxidative Stress Research Lab, Institute of Biological Sciences, Federal University of Para, Belem 66075-110, PA, Brazil; evertonlpvarela@gmail.com (E.L.P.V.); percario@ufpa.br (S.P.); 3Postgraduate Program in Pharmaceutical Sciences, Institute of Health Sciences, Federal University of Para, Belem 66075-110, PA, Brazil; jorddynevescruz@gmail.com (J.N.C.); gleisonhist@gmail.com (G.G.F.); 4Postgraduate Program in Biodiversity and Biotechnology of the BIONORTE Network, Federal University of Para, Belem 66075-110, PA, Brazil; 5Evandro Chagas Institute, Belem 66093-020, PA, Brazil; liliane.carneiro@cenp.gov.br

**Keywords:** *Eleutherine plicata*, isoeleutherine, eleutherine, eleutherol, oxidative stress

## Abstract

From *Eleutherine plicata*, naphthoquinones, isoeleutherine, and eleutherol were isolated, and previous studies have reported the antioxidant activity of these metabolites. The present work evaluated the role of oxidative changes in mice infected with Plasmodium berghei and treated with *E. plicata* extract, fraction, and isolated compounds, as well as to verify possible oxidative changes induced by these treatments. *E. plicata* extracts were prepared from powder from the bulbs, which were submitted to maceration with ethanol, yielding the extract (EEEp), which was fractionated under reflux, and the dichloromethane fraction (FDMEp) was submitted for further fractionation, leading to the isolation of isoeleutherine, eleutherine, and eleutherol. The antimalarial activity was examined using the suppressive test, evaluating the following parameters of oxidative stress: trolox equivalent antioxidant capacity (TEAC), thiobarbituric acid reactive substances (TBARS), and reduced glutathione (GSH). Furthermore, the molecular docking of naphthoquinones, eleutherol, eleutherine, and isoeleutherine interactions with antioxidant defense enzymes was investigated, which was favorable for the formation of the receptor–ligand complex, according to the re-rank score values. Eleutherine and isoeleutherine are the ones with the lowest binding energy for catalase (CAT), glutathione reductase (GR), and glutathione peroxidase (GPx1), showing themselves as possible targets of these molecules in the involvement of redox balance. Data from the present study showed that treatments with *E. plicata* stimulated an increase in antioxidant capacity and a reduction in oxidative stress in mice infected with *P. berghei*, with naphthoquinones being responsible for reducing oxidative changes and disease severity.

## 1. Introduction

Malaria is one of the most serious infectious diseases, affecting low- and middle-income countries, where pregnant women and children under five years of age are among the population at risk. In 2020, there were 241 million new cases in 87 countries, resulting in 627,000 deaths [1]. There are still some concerns for the treatment of malaria, such as the limited therapeutic options for the treatment of malaria in pregnant women, especially in the first trimester. Another important threat is the increasing resistance of the parasite *Plasmodium* to available drugs [2]. Therefore, searching for newer, safer, and even more effective therapeutic options is of utmost necessity [2].

Historically, medicinal plants have always contributed to the treatment of malaria, from where two of the main drugs used for treatment were originally isolated, quinine [3] and artemisinin [4,5]. Plant species, such as naphthoquinone-rich *E. plicata*, exhibit antimalarial activity. These ethanol extracts were obtained and fractionated, and the isolated naphthoquinones, eleutherine, and isoeleutherine (Figure 1). Their in vitro activity against *Plasmodium falciparum* was examined, and eleutherin (IC50 = 10.45 µg/mL) and isoeleutherine (IC50 = 8.70 µg/mL) showed the best activity toward the parasite.

Through molecular docking, it was demonstrated that these compounds (isoeleutherine, eleutherine, and eleutherol) interact with highly conserved residues from the binding cavity of the cytochrome bc1 complex, a protein (in mitochondria [6]. Some authors suggest that the mechanism of action of naphthoquinones involves the formation of reactive oxygen species (ROS) induced by the bioreduction of the quinonic complex, resulting in the oxidative stress of cells or induction of apoptosis [7,8,9]. Based on this, our study proposes that oxidative stress is one of the feelings related to the biological activities of naphthoquinones present in *E. plicata*.

Furthermore, the in silico toxicity study suggests that isoeleutherine and eleutherine are toxic to algae and *Daphnia* spp., and extremely toxic to Medaka and Minnow fishes. Regarding cytotoxicity to hERG, isoeleutherine and eleutherine presented medium risk and appeared to be carcinogenic only for rats. In the comet assay, it was observed that isoeleutherine caused high DNA damage but was slightly lower than that by doxorubicin. The *Allium cepa* bioindicator test of genotoxicity demonstrated that isoeleutherine, at all concentrations evaluated, did not significantly interfere with the mitotic index (MI) and did not cause anomalies in the mitosis cycle, and no micronucleated cells were observed. However, the exposure time interfered with the aberration index, mainly at anaphase, with bridges, multinucleated cells, and mitotic irregularities. There was a pronounced increase in MI over 72 h with the treatment with eleutherine. The indices of aberrations caused by eleutherin at concentrations of 12.5 and 25.0 µg/mL were higher than the indices of isoeleutherine. Unlike isoeleutherine, cells treated with eleutherine (25 µg/mL) showed micronucleus, bud, and mitotic irregularities during metaphase. In contrast, at anaphase, changes similar to those caused by isoeleutherine were observed. These results reinforce the hypothesis that eleutherine has a greater toxic potential than isoeleutherine [9].

In a study of the antitumor activity of eleutherine isolated from *E. plicata*, it was observed that this naphthoquinone had cytotoxic effects capable of reducing the proliferation of C6 cells in a dose-dependent manner, as well as suppressing tumor migration and invasion. Furthermore, it reduced protein kinase B (AKT) phosphorylation and telomerase expression, which results in cell cycle disruption and induces apoptosis in glioma cells [10].

The genotoxicity of ethanol extract (EEEp), dichloromethane fraction (FDMEp), and isoeleutherine isolated from *E. plicata* were also evaluated, using the micronucleus test on human hepatoma cells (HepG2). The fractionation of EEEp contributed to the increase in frequency micronucleus. The FDMEp presented higher rates than EEEp and isoeleutherine with lower toxicity. Isoeleutherine and 22 analogues underwent toxicity prediction studies in different models, and structural alterations do not seem to change the toxicity. We selected CP13, CP14, CP17, and isoeleutherine to perform the TOPO II molecular docking study, which showed that the compounds have good complementarity in the active site with important hydrogen bonds. Therefore, the structural changes in isoeleutherine led to us obtaining a molecule with a lower mutagenic potential, and the CP13 can be considered a prototype compound for the development of new molecules with pharmacological potential [11].

Another assessment of cytotoxicity and genotoxicity was performed through the Comet assay (alkaline version), and acute and subacute oral toxicities were also evaluated for *E. plicata*. The antioxidant capacity of the samples was evaluated in the wells where the cells were treated with *E. plicata*. FDMEp and isoeleutherine were cytotoxic, with higher rates of DNA fragmentation observed, and all samples displayed higher antioxidant potential than the control. In the acute oral toxicity test, EEEp, FDMEp, and isoeleutherine did not cause significant clinical changes. In the subacute toxicity assay, EEEp and FDMEp also did not cause clinical, hematological, or biochemical changes. Moreover, isoeleutherine, eleutherine, and eleutherol (Figure 1) bound similarly to caspase-8 [12].

The present work evaluated the role of oxidative changes in mice infected with *P. berghei* treated with *E. plicata* extract, fraction, and isolated compounds, and the possible oxidative alterations caused by the different treatments were assessed.

## 2. Results and Discussion

### 2.1. Chemistry

The ethanol extract was fractionated, yielding four fractions of which the dichloromethane fraction was used for further fractionation, yielding the isolation of the isoeleutherine, eleutherine, and eleutherol. The entire isolation and identification process are described in the work by Castro et al. [6] since in both studies the same samples obtained from *E. plicata* were used.

### 2.2. Changes in Oxidative Stress in Animals Treated with Fractionated Extract and Isolated Compounds of E. plicata

Mice infected with *P. berghei* were submitted to treatment with EEEp, FDMEp, isoeleutherine, eleutherine, and eleutherol, and markers of oxidative stress in the animals’ blood were measured. For the trolox equivalent antioxidant capacity (TEAC) of blood samples of groups treated with EEEp, only the EEEp group 200 mg/kg showed a significant increase in antioxidant capacity in relation to the positive control (*p* < 0.0001; Figure 2a). For the FDMEp groups, only FDMEp doses 50 and 100 mg/kg showed a significant increase compared to the positive control (*p* = 0.0002 and *p* < 0.0001, respectively; Figure 2b). For groups treated with naphthoquinones alone, isoeleutherine dose 30 mg/kg (*p* = 0.0037; Figure 2c), eleutherine dose 45 mg/kg (*p* = 0.0009; Figure 2d) and eleutherol dose 15 and 45 mg/kg (*p* = 0.0030 and *p* = 0.0006, respectively; Figure 2e) showed a statistically significant increase in TEAC values compared to the positive control.

Regarding the total antioxidant capacity analyzed by the TEAC test (Figure 2), it was observed that the treatments with *E. plicata* increased the TEAC concentration in the animals. This suggests that treatments stimulate antioxidant defense as host parasitemia progresses in an attempt to combat oxidative stress generated by infection.

This hypothesis was verified by Moreira et al. [13], who identified an increase in TEAC after the 10th day of treatment with dexamethasone, followed by an increase in survival and reduction in parasitemia of mice infected with *P. berghei*. Similar results were obtained during treatment with N-nitro-L-arginine methyl ester (L-NAME), inhibitor of nitric oxide (NO) synthesis, with increased TEAC on the 20th day post *P. berghei* infection [14].

Gomes et al. [15] observed that supplementation with *Agaricus sylvaticus* and N-acetylcysteine in infected *P. berghei* mice significantly increases TEAC levels between the 7th and 10th day post infection, resulting in reduced parasitemia and oxidative stress generated by the disease. Thus, based on our results, it is possible that the treatments with naphthoquinones present in *E. plicata* stimulate the synthesis of endogenous antioxidant molecules proportionally to the oxidative aggression generated by the host, because TEAC is a nonspecific marker of antioxidant defenses.

During plasmodium infection, the antioxidant capacity of the body is responsible for reducing the oxidative damage generated by the disease, and both the enzymatic and non-enzymatic antioxidant defense, especially GSH, can be stimulated by drug treatments, extracts of medicinal plants, and antioxidant vitamins, with the aim of strengthening the body against the aggression of ROS [16].

Regarding the GSH values in the treated groups, both the EEEp (Figure 3a) and the FDMEp (Figure 3b) at the doses of 100 and 200 mg/kg presented elevated levels of GSH compared to the positive control (*p* < 0.0001). Regarding the groups treated with naphthoquinones alone, only isoeleutherine dose 15 mg/kg (Figure 3c) showed a significant reduction in GSH when compared to the positive control (*p* = 0.0340). However, if we compare the groups treated with the extracts (Figure 3a,b) and isolated compounds (Figure 3c–e) with the negative control group, it was observed that for all doses there was a statistically significant increase in GSH values.

GSH is a tripeptide (*ɣ*-L-glutamyl-L-cysteinyl-glycine) that acts in the detoxification of xenobiotic compounds, directly and indirectly, through the reduction in oxidizing species and as a cofactor in the synthesis of enzymes of the antioxidant defense system [17]. Its elevated levels are related to the increase in oxidative stress and significant improvement in the clinical evolution of pathologies [18,19].

Our results demonstrated increased GSH activity in mice treated with *E. plicata*, suggesting that the treatments increased oxidative stress to combat parasite-induced ROS. This effect is due to the process of reduction in electrophile compounds that, when reacting with GSH, form GSSG, its oxidized form, which has a disulfide bond [20]. Among the electrophile compounds, O^2−^• and OH• radicals are neutralized directly by GSH, and the lipid peroxides are reduced by the action of GPX that uses GSH as a co-factor, demonstrating the indirect action of the antioxidant amino acid [16,21]

According to Muller et al. [22], the increase in antioxidant capacity through the activity of GSH and glutathione reductase (GR) results in a cascade of redox reactions capable of bioactivating potent naphthoquinones with antimalarial action, representing a new alternative to combat malaria. Thus, it is possible that the naphthoquinones present in *E. plicata* are responsible for increasing antioxidant capacity in response to oxidative stress, as identified in our study.

In other similar studies, Adeoye and Bewaji [23], while investigating the effectiveness of baobab stem bark extracts (Bombacaceae) in murine malaria, identified that the treatment reduced tissue peroxidation, resulting in increased levels of GSH, catalase (CAT), and superoxide dismutase (SOD) activity, concluding that the treatment stimulates enzymatic activity against the increase in oxidative stress generated by *P. berghei*. In the study by Atanu et al. [24], treatment with extracts of *Alstonia boonei* and *Carica papaya* in mice infected with *P. berghei* resulted in increased activity of GSH and antioxidant enzymes (SOD, CAT, GPX, and GSH) in response to oxidative stress caused by the parasite.

In the determination of thiobarbituric acid reactive substances (TBARS) of the mice, for the EEEp (Figure 4a) and the FDMEp (Figure 4b) in all doses evaluated, we observed a significant reduction in oxidative stress values compared to the positive control (*p* < 0.0001). For the isolated compounds, a similar behavior was observed, in which isoeleutherine 15 mg/kg (*p* = 0.0173), 30 and 45 mg/kg (*p* < 0.0001; Figure 4c), eleutherine (*p* < 0.0001; Figure 4d), and eleutherol reduced TBARS levels (*p* < 0.0001; Figure 4e) at all doses tested compared to the positive control.

The assessment of thiobarbituric acid reactive substances aims to quantify the formation of secondary products of lipid peroxidation such as malondialdehyde (MDA), one of the most widely used oxidative stress biomarkers. Thus, oxidative damage in the body is estimated through the action of oxidizing species to cellular components such as cell membrane lipids [25]. Regarding TBARS levels in our study, it was observed that treatment with *E. plicata* reduced the oxidative stress generated by the infection. This suggests that treatments based on naphthoquinones stimulated an increased antioxidant defense, generating a reduction in oxidative damage in malaria.

In a study conducted by Vale et al. [26] in which the antimalarial activity of *Himatanthus articulatus*, a plant used in the treatment of malaria in the Brazilian Amazon, was evaluated, a significant reduction in TBARS levels in *P. berghei*-infected mice was found, suggesting that the treatment reduces inflammation through the reduction in NO and the antioxidant properties of the ethanolic extract of the plant.

The reduction in TBARS levels in murine malaria was also identified in the study by Dkhil et al. [27] who used extracts from *Indigofera oblongifolia* leaves for the synthesis of silver nanoparticles (AgNPs). In addition, the study found a reduction in NO levels and increased antioxidant capacity of the animals, through the elevation of GSH and CAT. Similar results for the reduction in TBARS and increase in SOD and CAT were found in the study by Omonkhua et al. [28], which evaluated the antimalarial, hematological, and antioxidant effects of the methanolic extract of *Terminalia avicennioides* in *P. berghei*-infected mice.

### 2.3. Molecular Docking

Molecular docking evaluated the interaction of isoeleutherine, eleutherine, and eleutherol on SOD1, CAT, glutathione peroxidase (GPx1), and GR enzymes. The results regarding the binding energies obtained for the interactions of eleutherol, eleutherine, and isoeleutherine with SOD1, CAT, GPx1, and GR enzymes are summarized in Table 1.

The re-rank score values demonstrate that the interaction of the three molecules with each of the enzymes is favorable for the formation of the receptor–ligand complex, indicating a potential capacity to inhibit the activity of these proteins. To better understand the binding mode of the complexes, we performed an analysis on the nature of the chemical interactions established at the binding site of each enzyme.

The compounds eleutherol, eleutherine, and isoeleutherine showed binding affinity with the site located at the interface of the SOD1 dimers. Figure 5 shows the electrostatic surface of the protein with each of the molecules at its binding site. This site is essentially hydrophobic in nature; however, there are electrostatic interactions such as hydrogen bonds that are established with the ligands.

The lateral portions of the monomers are the ones that exhibit regions with the highest negative potential; this may occur due to the greater presence of amino acids with a charged side chain. Figure 6 shows the residues with which the molecules are able to interact, and we also highlight the hydrogen bonds established with the SOD1 residues.

In the formed complexes, the three ligands are able to interact with residues belonging to the two monomers of SOD1. Eleutherol performed hydrogen interactions with Asp11A and Asn53A; the other interactions formed with Gly56A, Lys9A, and Lys9B are hydrophobic in nature. The mode of binding of eleutherine with the bonding pocket allowed the emergence of hydrogen bonds with Asn53A and Lys9B and hydrophobic interactions with Lys9A and Gly56A. Isoeleutherine formed a hydrogen bond with Asp11A and had hydrophobic interactions with Asn53A, Lys9A, and Lys9B. The interactions established with Asn53 have been reported by Richard et al. [29] as important for the inhibition of SOD1 activity.

The catalase binding site for the three molecules exhibited a highly positive potential, as can be seen in Figure 7.

Figure 8 shows the various residues involved in the interaction of catalase with eleutherol, eleutherine, and isoeleutherine.

The three molecules interacted with the His466, Trp186, Gln168, and Arg127 present at the protein’s binding site. The imidazole ring of His466 and the indole ring of Trp186 performed pi–pi-like interactions with the nucleus of the three molecules. The following hydrogen interactions were also found: eleutherol with Gln168, Phe200, and Arg127, and eleutherine with Gln168, Asn62, Arg127, and Asp128. The other interactions established with the ligands were hydrophobic in nature.

The three ligands showed affinity of interaction with the interface of the monomers that make up glutathione peroxidase. This binding cavity in the three systems presented slightly hydrophobic potential (Figure 9).

All molecules interacted with residues present in the two protein chains and were also able to establish hydrogen bonds (Figure 10). Eleutherol formed these links with Gln76A, Asn75A, Phe145B, and Asp142B; eleutherine formed hydrogen interactions with Asn75A, Gln76A, Thr147B, and Asp142B; and isoeleutherine also showed this type of interaction with the residues Asn75A, Gln76A, Thr147A, and Asp142B. Hydrophobic interactions of the pi–pi type were established between the imidazole ring of the histidines present at the site with each of the docked molecules. Similar interactions can also be observed between the benzene ring of phenylalanine and the molecules under study.

Eleutherol, eleutherine, and isoeleutherine were folded in the active cavity in the glutathione reductase (GR), which presented positive potential on its surface (Figure 11).

The interaction cavity of the molecules is located near the FAD binding site (thiol/disulphide redox active center) which has been described as important for enzymatic catalysis performed by GR [30,31]. The molecules have molecular bonding poses in which they establish van der Waals interactions and hydrogen bonds (Figure 12). Eleutherol formed hydrogen interactions with Tyr114, Arg37, Ala34, and Gly29, and eleutherine established these interactions with Ser30 and Leu33, whereas isoeleutherine formed these interactions with Gly55 and Ser30.

Our ligands interacted with Cys58, which in some studies presents covalent binding mode; thus it is an important amino acid for enzymatic activity [32]. The ligands showed very similar binding sites to each other, so the molecules form interactions with residues similar to each other. Pi–pi interactions were observed between the molecules and Tyr114, whereas hydrophobic interactions were also formed with other residues such as: Val59, Gly55, Thr336, Ile343, Arg37, Leu53, Ala34, and Arg347.

The mechanism of action of naphthoquinones (isoeleutherine, eleutherine, and eleutherol) on antioxidant defense enzymes has not been clarified, but in a similar study that used 1, 4-naphthoquinones, it was shown that GR within infected red blood cells is the enzyme responsible for triggering a redox reaction cascade capable of bioactivating the antimalarial effects of naphthoquinones [22]. In its reduced form, benzoyl metabolites could convert methemoglobin into indigestible hemoglobin, which interrupts development and generates plasmodium death at the trophozoite stage [33].

In addition, it is known that part of the mechanism of action of quinones involves the generation of ERTON induced due to bioreduction of the quinonic nucleus by specific enzymes and oxygen [34]. In addition, such compounds are capable of inducing apoptosis and inhibiting the topoisomerases of tumor cells [9,11].

Based on the in silico study, it is likely that the enzymes SOD, CAT, and GR are inhibited by the action of naphthoquinones, which would generate greater bioreduction of the quinoidic substrate by the enzyme NADPH, forming a large amount of semiquinone radical anion species (Q^−●^) in situ and, consequently, generating a high production of O^2−●^ and H_2_O_2_ to fight the parasite. However, due to this possible mechanism, quinones would also present cytotoxic behavior for cancer and normal cells [35,36].

Quinones can present dual behavior, acting as pro-oxidant in micromolar concentrations and antioxidant in sub-micromolar doses, as is the case of MitoQ and plastoquinones, whose actions are specific at the mitochondrial level [37]. Thus, quinones may present toxic or protective characteristics in different biological systems [38].

Our study proposes that the possible interaction of naphthoquinones with antioxidant defense enzymes regulates oxidative stress levels in experimental malaria, resulting in activity. It should be noted that these enzymes are not the main targets of naphthoquinones. In addition, it is possible that the antimalarial action of naphthoquinones (isoeleutherine, eleutherine, and eleutherol) is due to the target site Qo of the mitochondrial complex Bc1 of the parasite, which results in the blockade of the energy generation of the mitochondria, a mechanism similar to that of atovacone [39]. However, this hypothesis was not investigated in the present study; new methods are necessary, such as the evaluation of gene expression of antioxidant enzymes and mitochondria in future studies.

## 3. Materials and Methods

### 3.1. Plant Material, Extract, and Fractions

*E. plicata* bulbs were collected in the municipality of Tracuateua, located in the state of Pará, Brazil (BR 308, Lat. 1.1436°, Long. 46.9511°). Botanical identification was performed by Dr. Márlia Regina Coelho Ferreira, and the specimen was deposited in the Herbarium João Murça Pires (MG) of the Museu Paraense Emílio Goeldi (Belém, PA, Brazil), under the registration MG. 202631. Subsequently, the bulbs were washed in running water, cut into smaller pieces, and kept drying in an air circulation oven. After drying, they were ground in facade mills to obtain the powder, and this powder was subjected to maceration in 96° GL ethanol. Then, the extractive solution was concentrated in a rotary evaporator, originating in the EEEp. The EEEp was subjected to fractionation by request under reflux, caused in fractions with increasing polarity. Due to its richness in quinone compounds, which were monitored by thin-layer chromatography, the FDMEp underwent a new fractionation in an open chromatographic column with solvents of increasing polarity, forming in the isolation of isoeleutherine, eleutherine, and eleuterol, which passed through the process of purification by recrystallization with methanol, and were then identified by nuclear magnetic resonance as described by Castro et al. [9,40].

### 3.2. Animals and Origin

The project was submitted to the Committee on Ethics in the Use of Animals-CEUA/UFPA and approved under report no. 7464060618 (ID 001020). A total of 195 mice of the Mus musculus species, Balb C-An lineage, with body mass ranging from 25 to 30 g, 120 female mice, were used for antimalarial and oxidative stress evaluation.

The animals came from the vivarium of Instituto Evandro Chagas (IEC; Ananindeua, PA, Brazil) and were kept at the experimental vivarium of the Oxidative Stress Research Laboratory (LAPEO; Federal University of Pará—UFPA, Belém, PA, Brazil). The animals were housed in cages in groups of eight animals each, kept under controlled temperature conditions (25 ± 1 °C) and in an alternated cycle of 12 h of light/dark. Water and food was provided ad libitum during the experiments.

All procedures with animals were conducted within the norms of animal experimentation, in accordance with the ethical principles in experimentation, according to the Brazilian Society of Sciences in Laboratory Animals—SBCAL and in compliance with international regulations.

### 3.3. Antimalarial Activity

The strain of *P. berghei* ANKA preserved in the whole blood of mice was provided by IEC. The strain inoculum was prepared in a solution in 1 *×* 10^6^ parasitized red blood cells per 0.2 mL of complete medium (RPMI 1640-Sigma Aldrich, Cat # R5886 and 5% fetal bovine serum -Invitrogen, Cat # 12657029), administered 0.2 mL per mouse, intraperitoneally. The rodent plasmodium model used in the evaluation of compounds with blood schizonticidal activity was the four-day suppressive test [41,42]. The animals were randomly divided into 8 groups, as follows:Group I (Negative Control; N = 10 animals): The animals were inoculated with and received physiological saline solution (0.9%; 1 mL/100 g body weight; orally for 4 days).Group II (Positive Control; N = 10 animals): The animals were inoculated with P. berghei-infected erythrocytes and received physiological saline solution (0.9%; 1 mL/100 g body weight; orally for 4 days).Group III (Chloroquine Group; N = 10 animals): The animals were inoculated with P. berghei-infected erythrocytes and received treatment with chloroquine (30 mg/kg body weight; orally for 4 days).Group IV (EEEp Group); N = 30 animals): It consisted of three subgroups of 10 animals each that were inoculated with P. berghei-infected erythrocytes and treated orally for 4 days with the ethanolic extract of *E. plicata*, at doses of 50 mg, 100 mg, and 200 mg/kg animal weight.Group V (FDMEp Group; N = 30 animals): It consisted of three subgroups of 10 animals each that were inoculated with P. berghei-infected erythrocytes and treated orally for 4 days with the dichloromethane fraction of *E. plicata*, at doses of 50 mg, 100 mg, and 200 mg/kg animal weight.Groups VI, VII, and VIII (Group Isoeleutherine; Eleutherine, and Eleutherol; N = 30 animals each group): It consisted of three subgroups of 10 animals each that were inoculated with P. berghei-infected erythrocytes and treated orally for 4 days with isoeleutherine (15, 30, and 45 mg/kg), eleutherine (15, 30, and 45 mg/kg), and eleutherol (15 and 45 mg/kg).

The study period lasted 8 days for all groups and, at the end of the period, animals were anesthetized intraperitoneally with ketamine (1.5 µL/g of body weight) and xylazine (0.5 µL/g of body weight) and submitted to euthanasia by exsanguination (hypovolemia) to obtain whole-blood samples by cardiac puncture, which were stored at −70 °C for later analysis.

### 3.4. Evaluation of Oxidative Stress Parameters

#### 3.4.1. Determination of Total Antioxidant Capacity

The total antioxidant capacity was determined according to the TEAC. Trolox (6-hydroxy-2,5,7,8-tetramethylchromo-2-carboxylic acid; Aldrich Chemical Co 23881-3) is a potent water-soluble synthetic vitamin E antioxidant. The method proposed by Miller et al. [43] modified by Re et al. [44] under adapted conditions of temperature, proportions relative to reagents and measurement time, was used. The method is based on the ability of substances to eliminate the ABTS^•+^ radical cation, a blue-green chromophore with maximum absorption at 734 nm, which decreases its intensity in the presence of antioxidants, depending on the duration of the reaction, antioxidant capacity, and concentration in the sample, resulting in the formation of ABTS, which is colorless. The ABTS^•+^ solution (2.45 mM) was prepared from the reaction between ABTS (7 mM; Sigma-Aldrich; A1888; São Paulo, SP, Brazil) and potassium persulfate (140 mM; K2O8S2; Sigma-Aldrich; 216224; São Paulo/SP). Initially, the reading (T0) of the ABTS^•+^ solution was performed. Then, 30 μL of sample or standard were added to the solution, and after 5 min, the second reading (T5) was performed. Deionized water was used as a white control and Trolox was used as a positive control. The reaction was measured in an 800XI spectrophotometer (Femto; São Paulo, SP, Brazil) at 734 nm. The final results were expressed in micromoles per liter (µM/L) corresponding to the concentration of trolox with antioxidant capacity equivalent to the test sample, measurement standard called TEAC.

#### 3.4.2. Determination of Reduced Glutathione

The determination of intracellular levels of the reduced form of GSH is based on the ability of GSH to reduce 5,5-dithiobis-2-nitrobenzoic acid (DTNB; Sigma-Aldrich) to nitrobenzoic acid (TNB), which will be quantified via spectrophotometry at a wavelength of 412 nm. For the determination of GSH concentrations, the methodology adapted from Ellman [45] was used, which removes the red blood cell concentrate, resulting in an aliquot of 20 μL in a test tube and includes 20 μL of distilled water and 3 mL of distilled water/EDTA, so that the 1st reading of the sample is performed (T0); then, 100 μL of DTNB is added, and after 3 min, the 2nd reading of the sample is performed (T3), and the concentration of GSH is expressed in μg/mL.

#### 3.4.3. Determination of Thiobarbituric Acid Reactive Substances

It was performed according to the method proposed by Kohn and Liversedge [46], modified by Percario et al. [47]. This method evaluates lipid peroxidation and was used as an indicator of oxidative stress. The test is based on the reaction of thiobarbituric acid (4,6-dihydroxypyrimidine-2-thiol, TBA; Sigma-Aldrich; T5500; São Paulo, SP, Brazil) with byproducts of lipid peroxidation [such as molondialdehyde (MDA)], at acid pH (2.5) and elevated temperature (94 °C), forming chromogens with absorbance at 535 nm. Initially 0.5 mL of the sample or standard was mixed with 1mL of TBA solution (10 mM). Then, this solution was placed in a water bath at 94 °C for 60 min. Subsequently, 4 mL of n-butyl alcohol was added, and the solution was stirred on a vortex-type shaker and then centrifuged at 3000 rpm for 10 min. After this, 3 mL of the supernatant was transferred to a cuvette and then spectrophotometry was performed at 535 nm (Spectrophotometer 800XI; Femto; São Paulo, SP, Brazil), and then, compared to the MDA standard, 1,1,3,3-tetrahydroxypropane (Aldrich chemistry, Cat # T9889).

### 3.5. Molecular Docking and Investigation of Antioxidant Capacity

The molecular structure of eleutherol, eleutherine, and isoeleutherine was drawn with the GaussView 5 software and then optimized with B3LYP/6-31G* [48,49] using the Gaussian16 package [50]. To analyze the interaction mechanism of these structures with antioxidant enzymes, Molegro Virtual Docker 5.5 was used [51]. MolDock scoring functions (GRID) and the MolDock algorithm were used in molecular docking simulations. The receptor and linkers were prepared using a standard MVD preparation module. For each complex, hydrogen atoms were added, and the program’s standard partial atomic charges were employed. The docking calculations were performed using the MolDock scoring function, with a grid resolution of 0.3 Å. The crystallographic structure of the antioxidant proteins SOD, CAT, GPx, and GR used as receptors can be found in the Protein Data Bank (PDB) from PDB ID: 2C9V (SOD) [52]; 1QQW (CAT) [53]; 1GP1 (GPx) [54]; and 1XAN (GR) [55].

### 3.6. Statistical Analysis

Statistical analysis was performed using the Graph pad Prisma 5 program. For each analyzed parameter, the analysis of possible discrepant points (outliers) was performed, which uses the interquartile range in its calculation, with discrepant points not considered in the statistical calculations. For each analyzed parameter, the homoscedasticity of the dispersion was evaluated, applying the Variance Analysis test (ANOVA) for the homoscedastic dispersion. After the existence of significant differences, these were compared between groups using Tukey’s post hoc test. In all tests, a significance level of 5% was considered (*p* ≤ 0.05).

## 4. Conclusions

The study data showed that treatments with *E. plicata* stimulated an increase in mobilized antioxidant capacity and a reduction in oxidative stress in mice infected with *P. berghei*; and naphthoquinones isolated from this plant were responsible for reducing oxidative changes and disease severity. Furthermore, the molecular docking of naphthoquinones, eleutherol, eleutherine, and isoeleutherine interactions with antioxidant defense enzymes was favorable for the formation of the receptor–ligand complex, according to the re-rank score values. Eleutherin and isoeleutherine were the ones that presented the lowest binding energy for CAT, GR, and GPx1, showing themselves to be possible targets of these molecules in the involvement of the redox balance during malaria.

## Figures and Tables

**Figure 1 molecules-28-05557-f001:**
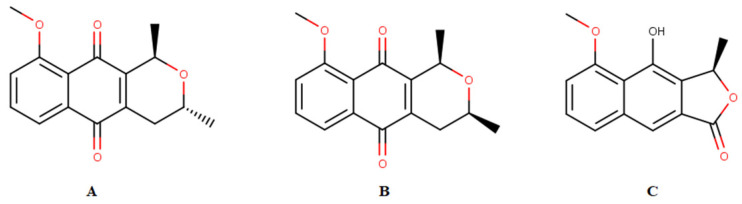
Chemical constituents isolated from *Eleutherine plicata*: (**A**) isoeleutherin, (**B**) eleutherol, and (**C**) isoeleutherine.

**Figure 2 molecules-28-05557-f002:**
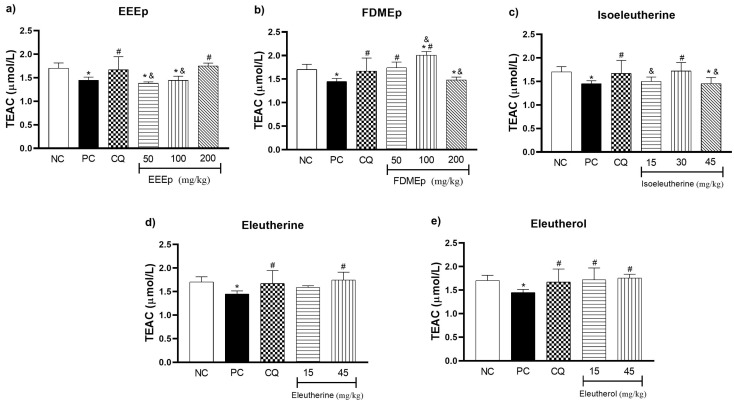
Levels of trolox equivalent antioxidant capacity (TEAC) in mice treated with *E. plicata*. (**a**) EEEp, (**b**) FDMEp, (**c**) isoeleutherine, (**d**) eleutherine, (**e**) eleutherol. Data correspond to means ± SD. * for all groups vs. negative control; # for all groups vs. positive control; & between treated groups; * *p* = 0.0012 positive control vs. negative control is the same result for all graphs; # *p* = 0.0074 chloroquine 30 mg/kg vs. positive control is the same result for all graphs. (**a**) * *p* < 0.0001 EEEp 50 mg/kg vs. negative control; * *p* = 0.0010 EEEp 100 mg/kg vs. negative control; # *p* < 0.0001 EEEp 200 mg/kg vs. positive control; & *p* < 0.0001 FDMEp 50 and 100 vs. 200 mg/kg; (**b**) * *p* < 0.0001 FDMEp 100 mg/kg vs. negative control; * *p* = 0.0098 FDMEp 200 mg/kg vs. negative control; # *p* = 0.0002 FDMEp 50 mg/kg vs. positive control; # *p* < 0.0001 FDMEp 100 mg/kg vs. positive control; & *p* = 0.0012 FDMEp 50 vs. 100 and 200 mg/kg; & *p* < 0.0001 FDMEp 100 vs. 200 mg/kg; (**c**) * *p* = 0.0105 isoeleutherine 45 mg/kg vs. negative control; # *p* = 0.0037 isoeleutherine 30 mg/kg vs. positive control; & *p* = 0.0321 isoeleutherine 15 mg/kg vs. isoeleutherine 30 mg/kg; & *p* = 0.0045 isoeleutherine 30 mg/kg vs. 45 mg/kg; (**d**) # *p* = 0.0009 eleutherine 45 mg/kg vs. positive control; (**e**) # *p* = 0.0030 eleutherol 15 mg/kg vs. positive control; # *p* = 0.0006 eleutherol 45 mg/kg vs. positive control; CN, negative control; CP, positive control, CQ, chloroquine 30 mg/kg.

**Figure 3 molecules-28-05557-f003:**
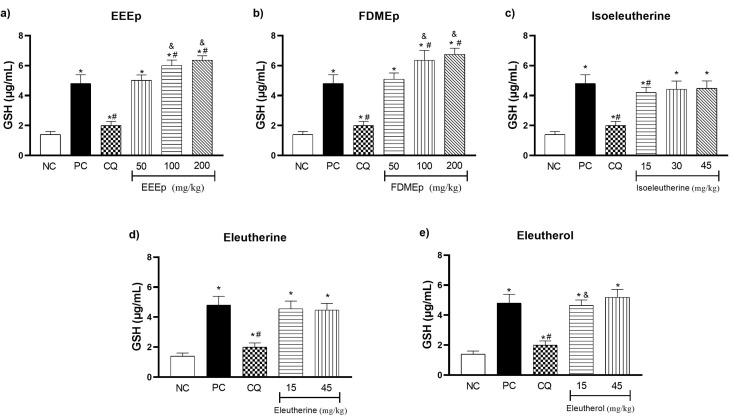
Levels of reduced glutathione (GSH) in mice treated with *E. plicata*. (**a**) EEEp, (**b**) FDMEp, (**c**) isoeleutherine, (**d**) eleutherine, (**e**) eleutherol. Data correspond to means ± SD. * for all groups vs. negative control; # for all groups vs. positive control; & between treated groups; * *p* = 0.0012 positive control vs. negative control is the same result for all graphs; # *p* = 0.0074 chloroquine 30 mg/kg vs. positive control is the same result for all graphs. (**a**) * *p* < 0.0001 EEEp 50, 100 and 200 mg/kg vs. negative control; # *p* < 0.0001 EEEp 100 and 200 mg/kg vs. positive control. & *p* < 0.0001 EEEp 50 vs. 100 and 200 mg/kg; (**b**) * *p* < 0.0001 FDMEp 50, 100 and 200 mg/kg vs. negative control; # *p* < 0.0001 FDMEp 100 and 200 mg/kg vs. positive control; & *p* < 0.0001 FDMEp 50 vs. 100 and 200 mg/kg; (**c**) * *p* < 0.0001 isoeleutherine 15, 30 and 45 mg/kg vs. negative control; # *p* = 0.0340 isoeleutherine 15 mg/kg vs. positive control; (**d**) * *p* < 0.0001 eleutherine 15 and 45 mg/kg vs. negative control; (**e**) * *p* < 0.0001 eleutherol 15 and 45 mg/kg vs. negative control; & *p* = 0.0433 eleutherol 15 vs. 45 mg/kg; CN, negative control; CP, positive control, CQ, chloroquine 30 mg/kg.

**Figure 4 molecules-28-05557-f004:**
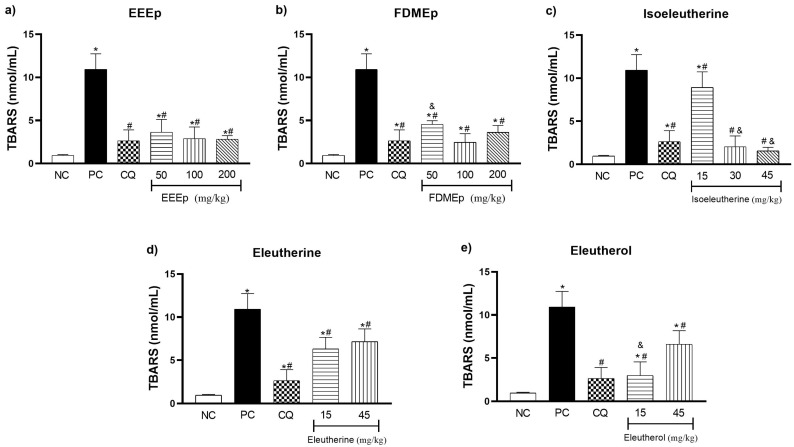
Levels of thiobarbituric acid reactive substances (TBARS) in mice treated with *E. plicata*. (**a**) EEEp, (**b**) FDMEp, (**c**) isoeleutherine, (**d**) eleutherine, (**e**) eleutherol. Data correspond to means ± SD. * for all groups vs. negative control; # for all groups vs. positive control; & between treated groups; * *p* = 0.0012 positive control vs. negative control is the same result for all graphs; # *p* = 0.0074 chloroquine 30 mg/kg vs. positive control is the same result for all graphs. (**a**) * *p* = 0.0002 EEEp 50 mg/kg vs. negative control; * *p* = 0.0130 EEEp 100 mg/kg vs. negative control; * *p* < 0.0191 EEEp 200 mg/kg vs. negative control. # *p* < 0.0001 EEEp 50, 100 and 200 mg/kg vs. positive control; (**b**) * *p* < 0.0001 FDMEp 50 mg/kg vs. negative control; * *p* = 0.0253 FDMEp 100 mg/kg vs. negative control; * *p* < 0.0001 FDMEp 200 mg/kg vs. negative control. # *p* < 0.0001 FDMEp 50, 100 and 200 mg/kg vs. positive control; & *p* = 0.0006 FDMEp 50 mg/kg vs. FDMEp 100 mg/kg; (**c**) * *p* < 0.0001 isoeleutherine 15 mg/kg vs. negative control; # *p* = 0.0173 isoeleutherine 15 mg/kg vs. positive control; # *p* < 0.0001 isoeleutherine 30 and 45 mg/kg vs. positive control; & *p* < 0.0001 isoeleutherine 15 mg/kg vs. isoeleutherine 30 and 45 mg/kg; (**d**) * *p* < 0.0001 eleutherine 15 and 45 mg/kg vs. negative control; # *p* < 0.0001 eleutherine 15 and 45 mg/kg vs. positive control; (**e**) * *p* = 0.0391 eleutherol 15 mg/kg vs. negative control. * *p* < 0.0001 eleutherol 45 mg/kg vs. negative control. # *p* < 0.0001 eleutherol 15 and 45 mg/kg vs. positive control. & *p* < 0.0001 eleutherol 15 vs. 45 mg/kg; CN, negative control; CP, positive control, CQ, chloroquine 30 mg/kg.

**Figure 5 molecules-28-05557-f005:**
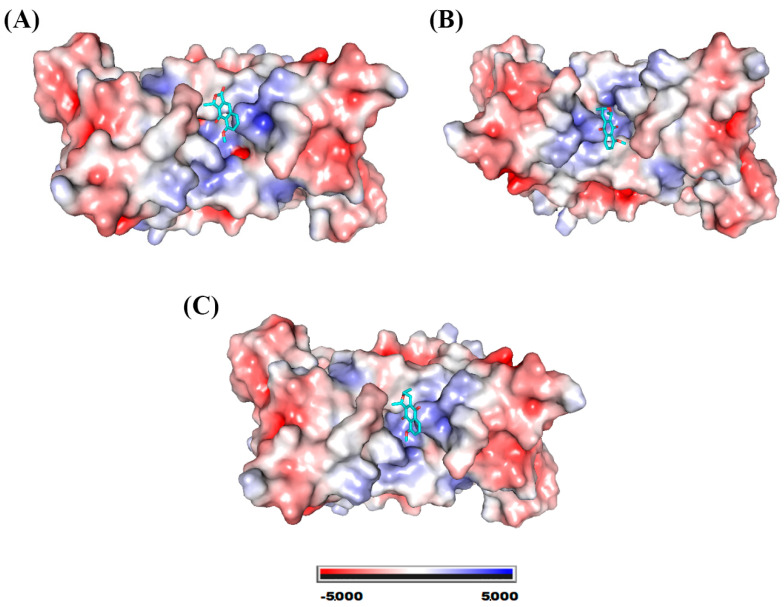
SOD1 electrostatic potential map. System formed with (**A**) eleutherol, (**B**) eleutherine, and (**C**) isoeleutherine.

**Figure 6 molecules-28-05557-f006:**
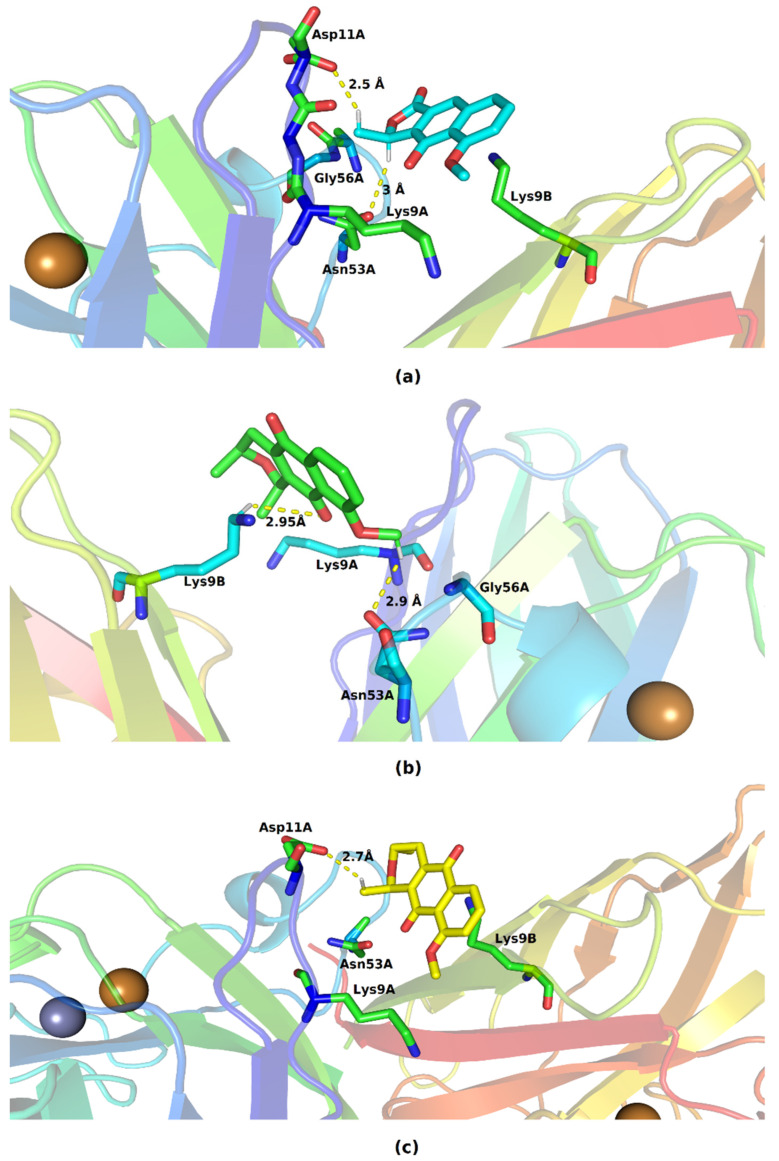
Key residues involved in the binding of (**a**) eleutherol, (**b**) eleutherine, and (**c**) isoeleutherine with SOD1. Distant residues up to 4 Å of the ligand are displayed on sticks, and hydrogen bonds are represented by lines dotted in yellow. Atom color code: oxygen atoms are colored red, nitrogen atoms blue, and hydrogen atoms white.

**Figure 7 molecules-28-05557-f007:**
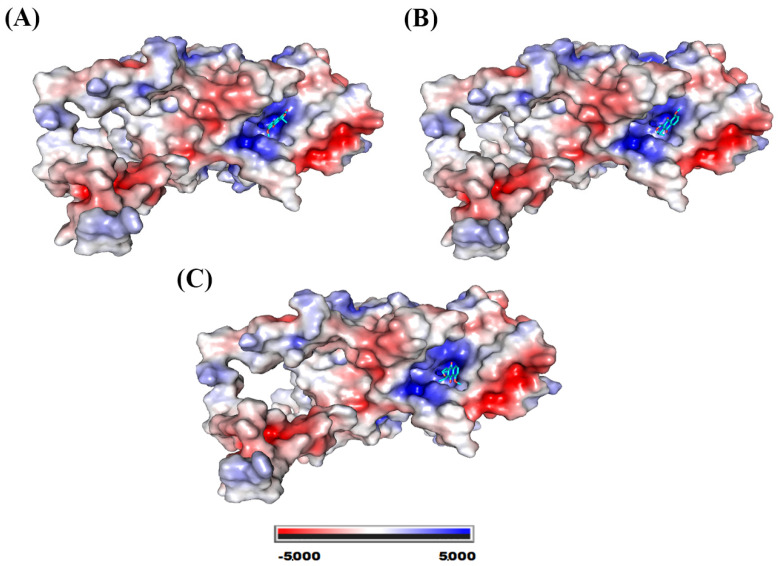
Catalase electrostatic potential map. System formed with (**A**) eleutherol, (**B**) eleutherine, and (**C**) isoeleutherine.

**Figure 8 molecules-28-05557-f008:**
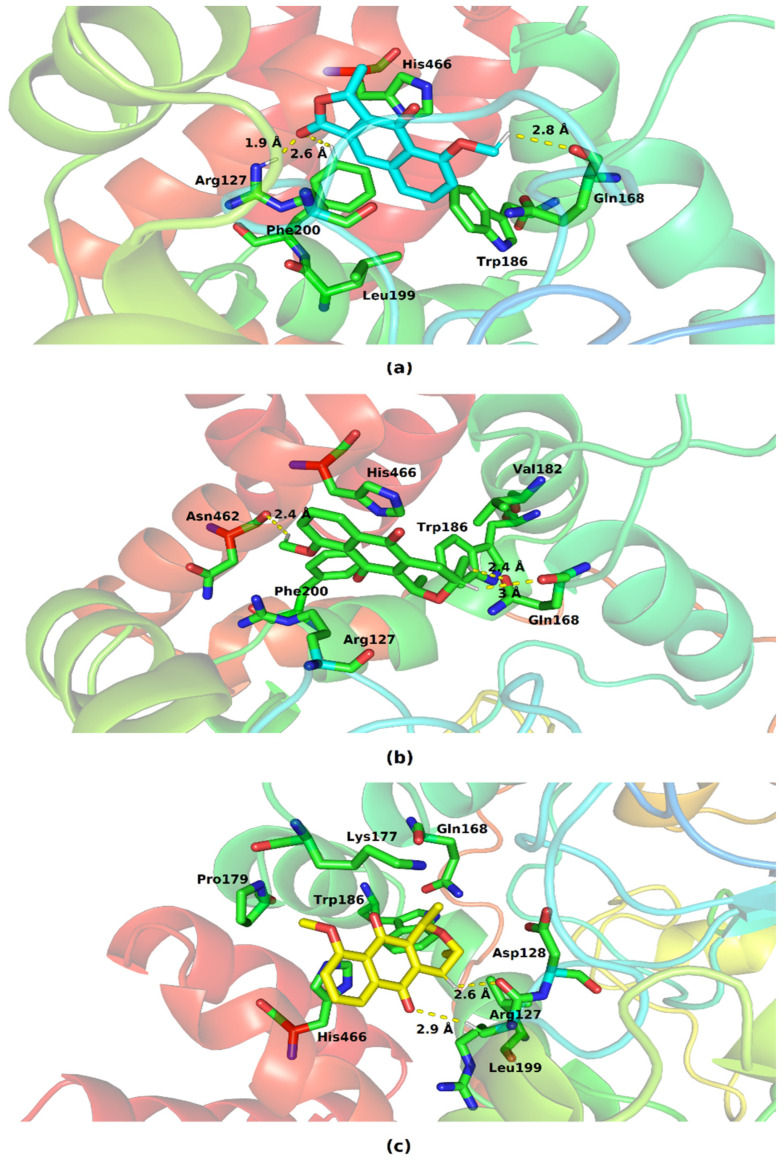
Key residues involved in the binding of (**a**) eleutherol, (**b**) eleutherine and (**c**) isoeleutherine with catalase. Distant residues up to 4 Å of the ligand are displayed on sticks, and hydrogen bonds are represented by lines dotted in yellow. Atom color code: oxygen atoms are colored red, nitrogen atoms blue, and hydrogen atoms white.

**Figure 9 molecules-28-05557-f009:**
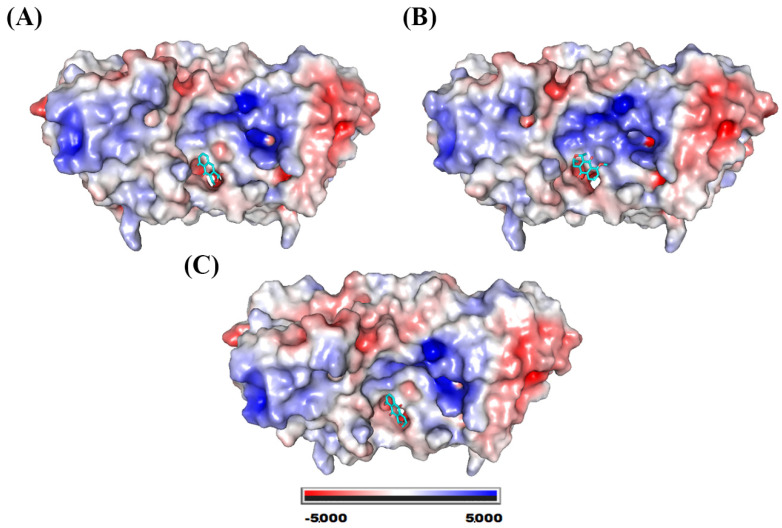
Electrostatic potential map of glutathione peroxidase. System formed with (**A**) eleutherol, (**B**) eleutherine, and (**C**) isoeleutherine.

**Figure 10 molecules-28-05557-f010:**
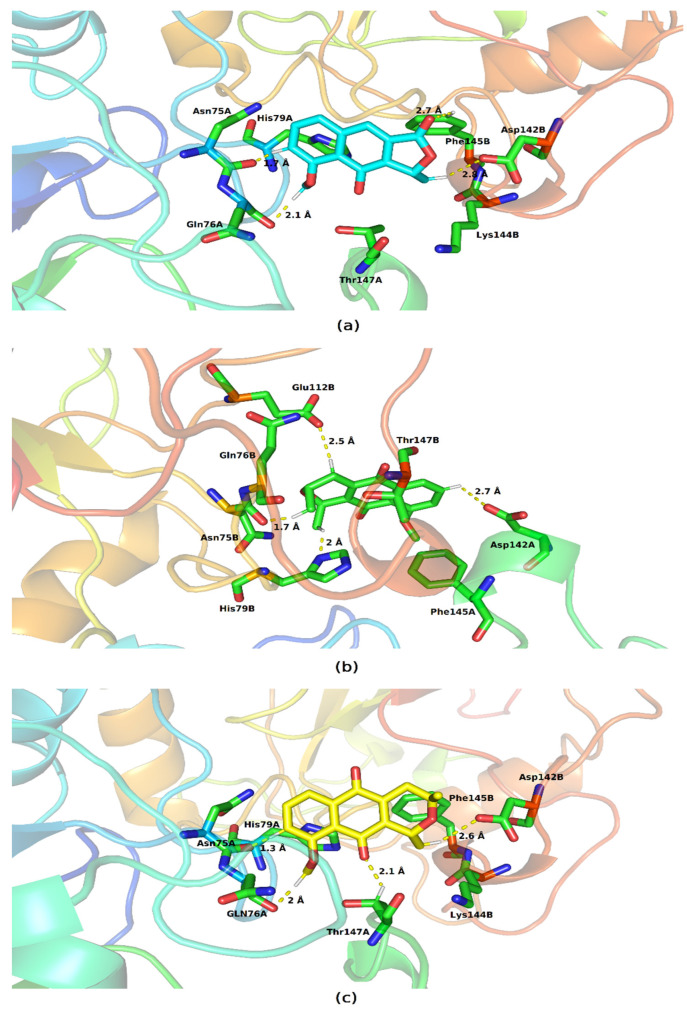
Key residues involved in the binding of (**a**) eleutherol, (**b**) eleutherine, and (**c**) isoeleutherine with glutathione peroxidase. Residues distant up to 4 Å from the ligand are displayed on sticks, and hydrogen bonds are represented by lines dotted in yellow. Atom color code: oxygen atoms are colored red, nitrogen atoms blue, and hydrogen atoms white.

**Figure 11 molecules-28-05557-f011:**
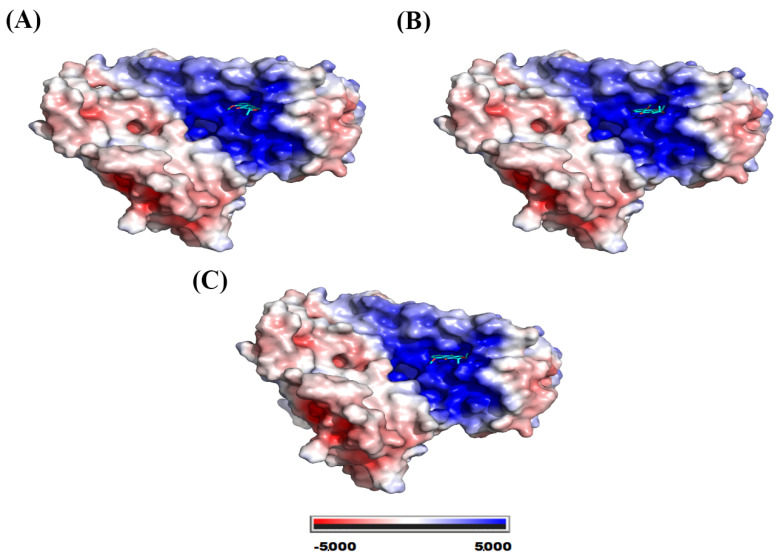
Electrostatic potential map of glutathione reductase. System formed with (**A**) eleutherol, (**B**) eleutherine, and (**C**) isoeleutherine.

**Figure 12 molecules-28-05557-f012:**
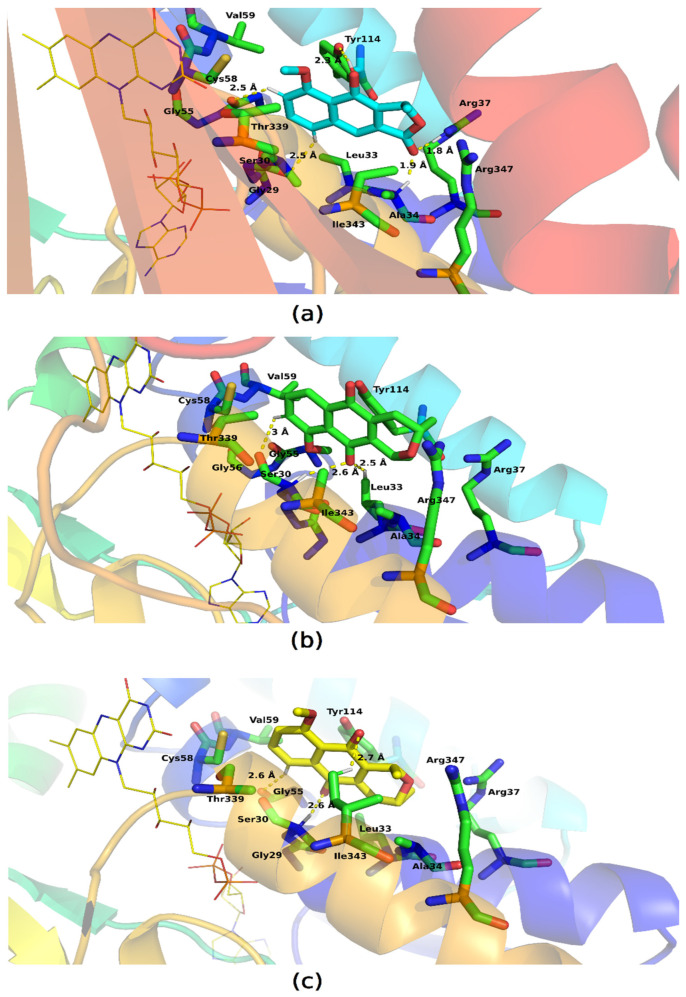
Key residues involved in the binding of (**a**) eleutherol, (**b**) eleutherine, and (**c**) isoeleutherine with glutathione reductase; the FAD is represented in yellow. Distant residues up to 4 Å of the ligand are displayed on sticks, and hydrogen bonds are represented by lines dotted in yellow. Atom color code: oxygen atoms are colored red, nitrogen atoms blue, and hydrogen atoms white.

**Table 1 molecules-28-05557-t001:** Binding energy of *E. plicata* compounds to enzymes involved in stress.

	System	Re-Rank Score
Eleutherol	Superoxide dismutase	−49.36
	Catalase	−64.36
	Glutathione peroxidase	−46.12
	Glutathione reductase	−69.97
Eleutherine	Superoxide dismutase	−37.63
	Catalase	−59.58
	Glutathione peroxidase	−27.08
	Glutathione reductase	−57.25
Isoeleutherine	Superoxide dismutase	−42.52
	Catalase	−57.42
	Glutathione peroxidase	−29.93
	Glutathione reductase	−57.07

## Data Availability

Data are available from the corresponding author upon request.

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
