# Peer review of "Participation of Oxidative Stress in the Activity of Compounds Isolated from Eleutherine plicata Herb"

_molecules, 2023, doi:10.3390/molecules28145557_

Round 1

Reviewer 1 Report

For references, perhaps they should be supplemented with the 2022 paper by Shinkai et al.  I think the wording "antiplasmodial activity" and "antimalarial activity" should be removed because the only antioxidant activity is proven. In subchapter 3.4.2., line 487, the title begins with a capital letter.

Author Response

For references, perhaps they should be supplemented with the 2022 paper by Shinkai et al.  I think the wording "antiplasmodial activity" and "antimalarial activity" should be removed because the only antioxidant activity is proven. In subchapter 3.4.2., line 487, the title begins with a capital letter.

Regards reviewer,

The wordings "antiplasmodial activity" and "antimalarial activity" have been removed from the text and the title, yet the reference has been added to the introduction and the item has been corrected.

We appreciate your cooperation.

Reviewer 2 Report

The research topic is interesting and relevant to public health since it intends to contribute to the search for alternative therapeutic agents for malaria. In addition, it is evident that the study is a continuation of other investigations developed by the authors and represents an advance on the subject. However, for their publication, the authors must improve some critical aspects, such as:

1) Introduction: The wording needs to be improved. The background information must be presented in an organized manner and supported by bibliographical references. There are many missing citations, making it difficult to verify the information. For example, paragraph 3 (lines 51-56) lacks citations. In other cases, the citations are wrong; reference [6] does not correspond to experiments performed with the cytochrome bc1 complex. The introduction should describe the scope of current research and the extent to which previous studies have addressed the problem, noting where there are gaps that the present study aims to address.

2) Methodology. Describe in detail each methodology used; this will allow other researchers to replicate the experiments and validate the results.

- In the "Plant Material, Extract and Fractions" section, explain the purification procedure for the compounds (isoeleutherine, eleutherine, and eleutherol). How is the chemical characterization of these compounds made? Authors must correctly cite the publication that supports these protocols.

- Briefly present the description of antioxidant activity protocols. What was the treatment of the blood samples before each trial?

- Antioxidant activity, TEAC, GSH, and TBA assays must also be written in detail, indicating, for example, volumes of reactants, sample volume, reaction conditions, and reaction time; description of the blank, controls, and number of replicates.

- Most of the bibliographical citations of the methodology are not found in the manuscript. In the section, you reach reference 38, and methodology has 54 references.

-Regarding the design of the experiment, How many replicates were made of each treatment? How was the analysis done statistics of the results? What were the significance tests applied? What was the hypothesis behind the tests?

3. Results and Discussion

-Include the significance values (p-values) in the context of the presentation of the results; it is better to do it here than in the figure legend.

-In the legends of figures 2, 3, and 4, you can write something like: "Data correspond to means ± SD, calculated using "XX" replicates; treatments sharing the same character are not significantly different at "XX" % confidence".

-Correct the citations of the bibliographical references; most of them are wrong. This error makes manuscript analysis very difficult.

-Improve the quality of bar charts. I suggest using a larger graphic size.

-Perform a critical analysis of your results. Propose other methods to confirm your findings in future studies.

Minor editing of English language required

Author Response

Dear Reviewer,

First, we would like to thank you for your contributions to our work, as follows:

1) Changes were made to the structure of the paragraphs and a new reference was added. All references were double-checked.

2) Changes and descriptions in the methodology were made.

3) Information has been added to the text and references have been re-evaluated and double-checked.

The text was again read by a native English speaker and revised.

Kind regards.